# Sensitivity analysis of numerical modeling input parameters on wind turbine loads in deterministic transient load cases

Will Wiley<sup>1</sup>, Jason Jonkman<sup>1</sup>, and Amy Robertson<sup>1</sup>

<sup>1</sup>National Renewable Energy Laboratory, 15013 Denver West Parkway, Golden, CO 80401, USA

Correspondence: Will Wiley (will.wiley@nrel.gov)

#### Abstract.

Aero-hydro-elastic-servo numerical models used to design and analyze wind turbines are based on thousands of variable input parameters that dictate the inflow, aerodynamic, structural, and control characteristics of the system as well as sea state, hydrodynamic, and mooring characteristics for fixed-bottom and floating offshore wind turbines. Each of these parameters has some level of uncertainty, which can significantly impact the predicted loads. Understanding the uncertainty in the inputs is critical to understanding the uncertainty in the outputs. This work demonstrates a screening technique to identify which parameters ultimate loads are most sensitive to so that more focus can be given to quantifying the possible range of those parameters. This technique has been demonstrated previously for different turbine and load case types and is extended here for a floating offshore wind turbine in design load cases with transient events both in the inflow and operations. Each load case features a deterministic gust including variations in wind speed, direction, and shear. Load cases are considered with an operating turbine as well as with prescribed fault, startup, and shutdown procedures. The study found that key input parameters with a large impact on loads include the length of the gust, the magnitude of direction change and speed in the gust, the initial wind speed, and the shape of the gust profile.

Copyright statement. This work was authored by NREL for the U.S. Department of Energy (DOE) under contract no. DE-AC36-08GO28308. Funding provided by U.S. Department of Energy Office of Energy Efficiency and Renewable Energy Wind Energy Technologies Office. The views expressed in the article do not necessarily represent the views of the DOE or the U.S. Government. The U.S. Government retains and the publisher, by accepting the article for publication, acknowledges that the U.S. Government retains a nonexclusive, paid-up, irrevocable, worldwide license to publish or reproduce the published form of this work, or allow others to do so, for U.S. Government purposes.

#### 1 Introduction

Transient conditions and operations can lead to design-driving ultimate loads for wind systems, including land-based wind turbines and both fixed-bottom and floating offshore wind turbines. The International Electrotechnical Commission (IEC) prescribes design load cases that include a number of transient events to ensure that structures can withstand loads in these scenarios. Transient load cases include gusts, faults, startup, and shutdown (IEC, 2024). Engineering-fidelity aero-hydro-servo-

https://doi.org/10.5194/wes-2025-228

Preprint. Discussion started: 14 November 2025

© Author(s) 2025. CC BY 4.0 License.

30

elastic codes can provide valuable predictions of the complex highly coupled loads that wind turbines encounter. These tools are based on models that represent the inflow wind, aerodynamic loading and flow disturbance, structural motions and deformations, meteorological and oceanographic (metocean) conditions, hydrodynamic loading, mooring systems, and turbine controllers. Modeling each part of the system requires many different input parameters that define the device and physical phenomena that impact it. Each input parameter has some range of uncertainty stemming from the stochastic nature of the environment, variability in manufacturing, changes that occur over the life of a project, changes in the surrounding environment that impact flow conditions, lack of available information, or combinations of these. Variations in some input parameters may have large impacts on certain load predictions while variations in other input parameters may not result in significant changes. Understanding this sensitivity is critical for probabilistic design and further investigation into the uncertainty of the parameters that drive design loads.

An elementary effects (EE) approach has been used in many previous wind turbine studies to identify the sensitivity to variability of input parameters that are most important for ultimate and fatigue load prediction. These studies focused on a range of system characteristics across different turbines and support structures, including turbulent inflow wind (Robertson et al., 2018), airfoil properties (Shaler et al., 2019), wind and turbine parameters (Robertson et al., 2019), farm and wake characteristics (Shaler et al., 2023), inflow and turbine parameters for a floating system (Wiley et al., 2023; Reddy et al., 2024), monopile fatigue properties (Sørum et al., 2022), and inflow, sea state, and turbine parameters in extreme storm load cases (Wiley et al., 2025). All of the mentioned previous works focused on stochastic definition of the environment characterized by some statistical quantities and a turbine condition to match the environment. The present study extends the demonstrated EE technique to investigate the sensitivity specifically in transient load cases. The tested load cases combine a deterministic gust with a turbine in normal operation, a loss of electrical system fault, a startup procedure, and a shutdown procedure. Numerical modeling is performed with the extensively used open-source code OpenFAST (NREL, 2024).

#### 45 2 Transient load cases

Table 1 shows the definition of the relevant IEC design load cases with deterministic gust inflow conditions, which are known from 30 years of industrial wind turbine design experience to be potentially design-driving. The load cases include four load case types, or "design situations": power production (OP), fault (FT), startup (SU), and shutdown (SD). These load cases are focused on identifying important ultimate loads and are assigned different gust types and partial safety factors.

© Author(s) 2025. CC BY 4.0 License.

**Table 1.** IEC transient load cases including gust. (DLC = design load case, ECD = extreme coherent gust with direction change, EWS = extreme wind shear, EOG = extreme operating gust, EDC = extreme direction change, Vr = rate of wind speed, Vin = cut-in wind speed, Vout = cut-out wind speed, NSS = normal sea state, MIS = misaligned, MUL = multidirectional, COD = co-directional, NCM = normal current model, MSL = mean sea level, U = ultimate strength, N = normal, A = abnormal)

| Design situation | DLC | Wind condition                                                                                                                         | Waves           | Metocean directionality | Sea currents | Water level | Other conditions                | Type of analysis | Partial safety factor |
|------------------|-----|----------------------------------------------------------------------------------------------------------------------------------------|-----------------|-------------------------|--------------|-------------|---------------------------------|------------------|-----------------------|
| Power production | 1.4 | ECD                                                                                                                                    | NSS             | MIS, wind direction     | NCM          | MSL         |                                 | U                | N (1.35)              |
| (OP)             | 1.4 | Vhub = Vr - 2  m/s, Vr, Vr + 2  m/s                                                                                                    | Hs = E[Hs Vhub] | change, MUL             | IVCIVI       | WISL        |                                 | O                | 14 (1.55)             |
| (01)             |     | EWS                                                                                                                                    | NSS             |                         |              |             |                                 |                  |                       |
|                  | 1.5 | Vin <vhub <vout<="" td=""><td>Hs = E[Hs Vhub]</td><td>COD, MUL</td><td>NCM</td><td>MSL</td><td></td><td>U</td><td>N (1.35)</td></vhub> | Hs = E[Hs Vhub] | COD, MUL                | NCM          | MSL         |                                 | U                | N (1.35)              |
| Fault            |     | EOG                                                                                                                                    | NSS             |                         |              |             | External or internal electrical |                  |                       |
|                  | 2.3 |                                                                                                                                        |                 | COD, MUL                | NCM          | MSL         | fault including loss of         | U                | A (1.1)               |
| (FT)             |     | Vhub = $Vr \pm 2$ m/s and Vout                                                                                                         | Hs = E[Hs Vhub] |                         |              |             | electrical network              |                  |                       |
| Startup          | 3.2 | EOG                                                                                                                                    | NSS             | COD, MUL                | NCM          | MSL         |                                 | U                | N (1.35)              |
| (SU)             | 3.2 | Vhub = Vin, Vr $\pm~2$ m/s and Vout                                                                                                    | Hs = E[Hs Vhub] | COD, MOL                | NCIVI        | MSL         |                                 | O                | N (1.33)              |
| (50)             |     | EDC                                                                                                                                    | NSS             | MIS, wind direction     | vev          | 1.00        |                                 | ••               | N (1.25)              |
|                  | 3.3 | Vhub = Vin, Vr $\pm$ 2 m/s and Vout                                                                                                    | Hs = E[Hs Vhub] | change, MUL             | NCM          | MSL         |                                 | U                | N (1.35)              |
| Shutdown         | 4.2 | EOG                                                                                                                                    | NSS             | COD, MUL                | NCM          | MSL         |                                 | U                | N (1.35)              |
| (SD)             | 7.2 | Vhub = $Vr \pm 2m/s$ and Vout                                                                                                          | Hs = E[Hs Vhub] | COD, MOL                | 110111       | MOL         |                                 | C                | 11 (1.55)             |

Inspired by these IEC design load cases, the present sensitivity study attempts to generalize and combine these load cases for the large number of simulations needed in the analysis sampling. The four load case types are run for the wind speeds shown in Table 2, for a total of 10 load case type and wind speed combinations. The selected load cases cover all relevant combinations of wind speeds and events from the IEC definitions in Table 1.

**Table 2.** EE sensitivity study load case type and wind speed combinations. Notations in parenthesis used as labels in following plots and tables.

|                |                  | Wind Speed      |                   |
|----------------|------------------|-----------------|-------------------|
|                | Near cut-in (VI) | Near rated (VR) | Near cut-out (VO) |
| Operating (OP) | X                | X               | X                 |
| Fault (FT)     |                  | X               | X                 |
| Startup (SU)   | X                | X               | X                 |
| Shutdown (SD)  |                  | X               | X                 |

#### 2.1 Parametric gust

The IEC design load cases include four types of deterministic gusts: extreme operating gust (EOG), extreme direction change (EDC), extreme wind shear (EWS), and extreme coherent gust with direction change (ECD). The inclusion of the ECD, which has a concurrent change in speed and direction, indicates that different descriptions of the wind are likely to change at the same time with the arrival of a gust. Field measurements also indicate that speed, direction, and shear often change together. In order to understand the relative importance of each of these components of a gust, a combined parametric gust is proposed and tested in this study.

The parametric gust is defined by seven input parameters. Four parameters define the magnitude of the changes in the gust: speed  $(GM_{speed})$ , direction  $(GM_{dir})$ , vertical shear  $(GM_{vshear})$ , and horizontal shear  $(GM_{ushear})$ . One parameter defines the

period of the gust, and two parameters define the shape of the gust. All four quantities use the same period and shape for a given simulation.

The two shape parameters,  $f_{return}$  and  $f_{dip}$ , describe whether the value returns to its pre-gust level after the gust peak and whether there is a dip in the opposite direction at the start and end of the gust. In the IEC gust definitions, the EOG and EWS return but the EDC and ECD do not, and the EOG dips but the EWS, EDC, and ECD do not. These are introduced as variable functions with a value of 1.0 following the IEC profile for a return or dip and a value of 0.0 following the IEC profile for no return or dip. Figure 1 shows the extents of what these parameters can look like over the length of the gust period.

Figure 1. Dip and return shape factors for parametric generalized gust profile

#### 70 2.2 Test system

The test system for the study is the IEA 15 MW reference wind turbine on top of the University of Maine VolturnUS-S semisubmersible platform. The 240 m diameter turbine was designed in 2020 under IEA Wind Task 37 and was meant to be a public reference representative of the growing size of large flexible modern offshore turbines (Gaertner et al., 2020). The VolturnUS-S was also designed under IEA Wind Task 37 and was intended to be representative of industry floaters (Allen et al., 2020). It features submerged rectangular pontoons in a "Y" configuration and three outer surface piercing columns with one central column supporting the tower. The turbine and platform are shown in Fig. 2.

**Figure 2.** University of Maine VolturnUS-S reference semisubmersible platform with the IEA 15 MW offshore reference wind turbine (Allen et al., 2020)

The same floating offshore wind turbine system was the subject of a 2025 EE sensitivity study focusing on extreme storm conditions (Wiley et al., 2025). Similar to this previous work, two locations are considered in the present study, Humboldt Bay and the Gulf of Maine. Two reference mooring systems were designed by the National Renewable Energy Laboratory (NREL) for these sites. A taut-chain-polyester-chain system was designed for the 800 m water depth at Humboldt Bay and a chain-polyester semi-taut system was designed for the 200 m water depth in the Gulf of Maine (Lozon et al., 2025). Visualizations of the two mooring systems are shown in Fig. 3.

100

**Figure 3.** Semi-taut and taut mooring systems for the VolturnUS-S designed for Gulf of Maine (left) and Humboldt Bay (right). Power cable shown in figure is not included in this analysis. (Lozon et al., 2025)

The transient load cases inherently depend on the turbine's controller and protection system, both for the prescribed turbine events of a fault, startup, and shutdown and automated reactions to the rapidly changing inflow wind when a gust arrives. The controller properties were briefly considered as possible variable input parameters to include in the sensitivity study, but it was decided that variability in the selection of these values is not analogous to the variability/uncertainty ranges in the other input parameters described in Sect. 4. Controller parameters are set by the designer or operator and should have no fluctuations from the set values. The controller is a design choice, and there is no clear range of possible values. Nevertheless, while control strategies were not included in the sensitivity study, it is important to use logic representative of reality.

The NREL Reference Open-Source Controller (ROSCO) is used in this study. Automatic shutdown representing the action of the protection system is enabled in all load cases based on thresholds for collective blade pitch, yaw error, and generator speed. In the shutdown, the generator torque is decreased linearly to zero, and the blades are pitched linearly to a feathered angle of 90 degrees. The generator torque rate and blade pitch rate should be fast enough to avoid dangerous operations but slow enough to avoid large transient loads from an immediate loss of thrust and torque. A sweep of pitch and torque rates were tested for a scheduled shutdown coincident with gusts that included speed, direction, vertical shear, and horizontal shear changes, both at wind speeds near rated and near cut-out. Four outputs were used to determine the shut-down rates that best minimize loads: blade-root bending moment, blade-tip out-of-plane deflection, tower-base bending moment, and maximum fairlead tension. The results are shown in Fig. 4 for the near cut-out wind speed. Slower pitch rates generally led to higher blade loads and deflections, while faster pitch rates generally led to higher tower and mooring loads. Dependence on torque rates over the tested range was smaller, but slower torque rates generally led to higher blade-root bending moments and mooring tensions, while higher torque rates led to higher blade deflections and tower loads. An optimal torque and pitch rate was selected for all normal and protection shutdowns of 150 kN-m/s and 1.25 deg/s, as shown with a red "X" in the plots. For the same sweep of torque and pitch rates at rated wind speed, the sensitivity was negligible, with maximum loads and deflections all occurring before shutdown was initiated.

**Figure 4.** Blade-root bending moment, blade-tip out-of-plane deflection, tower-base bending moment, and maximum fairlead tension during normal shutdown at near-cutout wind speed with a range of shutdown torque and pitch rates. Selected rates indicated with red X.

Startup procedures included a preliminary "safe power" hold before ramping up to full power. The scheme is shown in Fig. 5. A gradual ramp is used up to the reduced safe power. Speed and torque are then held for some time to ensure systems are working properly. There is then a ramp up to full operating power for the given wind conditions.

Figure 5. Startup procedure with initial safe power hold and ramp up

The IEC fault type prescribed in DLC 2.3 calls for a loss of electrical network (IEC, 2024). Many modern wind turbines have a battery backup system, which enables blade pitch maneuvers when grid connection is lost. To fit this relevant scenario, the fault case tested includes an immediate loss of generator torque followed by a blade pitch-to-feather with the maximum available blade pitch rate.

Table 3 provides the selected logic and parameters used for the study.

**Table 3.** Control properties for normal shutdown (also used for protection shutdown), startup, and loss of electrical network fault type.

| Shutdown (normal and protection)                 | Startup                         | Fault                                      |
|--------------------------------------------------|---------------------------------|--------------------------------------------|
| Decrease to no torque at optimal rate 150 kN-m/s | Safe power<br>20 % torque       | Immediate loss of torque                   |
| Pitch to feather at optimal rate 1.25 deg/s      | Safe power ramp time 100.0 s    | Pitch to feather at maximum rate 2.0 deg/s |
|                                                  | Safe power hold time<br>200.0 s |                                            |
|                                                  | Full power ramp time 100.0 s    |                                            |

For all runs, an initial simulation transient before either the turbine event or gust starts and an ending transient after the turbine event or gust ends were included. The initial transient had a length of three platform surge periods (429.0 s), and the end transient had a length of five platform surge periods (715.0 s). The relative timing of the turbine event (fault, startup, or shutdown) and the gust could include any possible overlap. Figure 6 shows this full range with some generic event and gust length. The event at the earliest could end just as the gust arrives, and at the latest, it could start just as the gust ends. This relative timing is assigned randomly with the seed number for each simulation.

Figure 6. Possible range of phase for event relative to gust. Relative timing is set randomly as a function of seed number.

The seed number is also used to set the phasing for the irregular wave field and to randomly set the initial azimuth of the rotor. All changes of the seed number add variation in the alignment of the motion of the system compared to the gust.

#### 3 Elementary effects

130

The EE method aims to identify the most important parameters with a relatively low number of required simulations. It was originally developed for general computational experiments by Max Morris in 1991 (Morris, 1991). The EE sampling method does not allow any identification of coupling between parameters. Robertson et al. (2018) laid out a modification of the method with radial perturbations of all parameters for use with engineering-fidelity wind turbine modeling. Many groups since have used this same method to focus on different components of a wind turbine system in different conditions.

Figure 7 shows an example simplified three-parameter EE sampling space. Each point in the figure represents a simulation run with a different set of parameters. The blue points are starting points placed throughout the parameter space, and the red points are perturbations in a single parameter from that starting point. The difference in predicted loads between the perturbation and the starting point is used to assess the local sensitivity to that parameter at that point in the parameter space. In this work

all perturbations are  $\pm 10$  % of the specific parameter range. The size of this perturbation is intended to result in identifiable changes in loads without smoothing out nonlinearities (Robertson et al., 2018). The direction of the perturbation (plus or minus) is randomly selected as a function of a random seed number. At the edges of the parameter space, the perturbation direction is constrained to ensure all simulations are within the limits of the selected parameter ranges (e.g., only negative perturbations for starting points near the upper boundary).

Figure 7. Visual representation of a three-dimensional EE parameter space

As more starting points are added to fill out the parameter space, the calculated sensitivity can approach the global sensitivity. The locations of the starting points were selected following a Sobol sequence, with the goal of a uniformly sampled parameter space. The advantage of the Sobol sequence is that additional points can be added to fill in gaps in the space without having to start over to add new samples, allowing the convergence of the global sensitivity to be tracked as additional points are added (Sobol, 1967).

In this study, all load cases focus on ultimate loads. Equation 1 shows the selection of the ultimate load for a given quantity of interest, Y, based on a simulation with a set of input parameters, U. The partial safety factor, PSF, is applied here based on the load case type and the IEC definition in Table 1. The tracked ultimate load for each output is then just the maximum of the absolute value within the time series, once the initial transient has been removed.

$$145 \quad Y_{ult.}(U) = MAX(PSF|Y(U)|) \tag{1}$$

The ultimate elementary effects value, UEE, is calculated following Eq. (2). UEE is a local partial derivative with the difference between the output calculated with the perturbation,  $x_i$ , and without the perturbation for that starting point,  $U^b$ , divided by the change in the specific input,  $\Delta_{iLC}$ . In order to compare the sensitivity across different input parameter types, this partial derivative is multiplied by the total range for that parameter,  $u_{iLC,range}$ . The sensitivity is then inherently dependent on the selection of the range, and this should be considered when assessing the outputs. Thus, the EE value depends not only on the gradient of output with respect to the input, but also the range of that input. It is in fact the multiplied combination of these two terms that is indicative of sensitivity. The absolute value of this change in outputs is then averaged across all seed numbers (NS is the total number of seeds simulated). Finally, the nominal output for the specific load case,  $\overline{Y_{LC}}$ , is added. For ultimate load assessment, only the largest value matters; the addition of this nominal output means that high sensitivity in a load case with low loads is not treated the same as a high sensitivity in a load case with larger loads.

https://doi.org/10.5194/wes-2025-228 Preprint. Discussion started: 14 November 2025 © Author(s) 2025. CC BY 4.0 License.

$$UEE_{iLC}^{b} = \frac{1}{NS} \sum_{s=1}^{NS} \left| \frac{Y_{ult.}(U^{b} + x_{i}) - Y_{ult.}(U^{b})}{\Delta_{iLC}} u_{iLC,range} \right| + \overline{Y_{LC}}$$

$$(2)$$

To identify the input parameter perturbations that are leading to the most significant changes in loads, some criterion is set to signify a "significant" EE value. This threshold is defined in Eq. (3), as any EE value that is greater than 2 standard deviations above the mean of all EE values.

$$160 \quad SEE > 2\sigma + \mu \tag{3}$$

Once significant EE events are identified, the sensitivity of a certain output to a certain input is determined by the relative number of significant events.

#### 4 Parameters and quantities of interest

The number of required simulations scales directly with the number of input parameters considered. It is helpful to only include variables that are expected to potentially lead to large or interesting sensitivities. Table 4 lists the 37 parameters considered in this study. The set of parameters is similar to those used with the same system in the 2025 analysis focusing on extreme storm conditions (Wiley et al., 2025). Parametric gust values replace any descriptions of turbulence. Unsteady aerodynamics parameters used in the dynamic stall model and detailed polar controls, previously found to lead to small sensitivities in normal operation, were considered again, with the expectation that they could be more important for transient conditions with rapidly changing angles of attack (Shaler et al., 2019; Robertson et al., 2019). Parameters found to have a low relative importance in the storm study with no justified expectation for increased relevance in transient load cases were not included.

Table 4. Parameter names and units.

| Parameter                                          | Name               | Range Units                |
|----------------------------------------------------|--------------------|----------------------------|
| Mean wind speed (before gust)                      | $U_{mean}$         | m/s                        |
| Shear exponent (before gust)                       | Shear              | -                          |
| Gust: magnitude of speed change                    | $GM_{speed}$       | m/s                        |
| Gust: magnitude of direction change                | $GM_{dir}$         | degrees                    |
| Gust: magnitude of vertical shear change           | $GM_{vshear}$      | m/s ( $\Delta$ over rotor) |
| Gust: magnitude of horizontal shear change         | $GM_{hshear}$      | m/s ( $\Delta$ over rotor) |
| Gust: period                                       | $T_{Gust}$         | s                          |
| Gust shape: return to initial value after the gust | $f_{return}$       | -                          |
| Gust shape: initial dip in opposite direction      | $f_{dip}$          | -                          |
| Lift coefficient at blade tip                      | $Cl_{tip}$         | $\Delta\%$                 |
| Drag coefficient at blade tip                      | $Cd_{tip}$         | $\Delta\%$                 |
| Lift coefficient at blade root                     | $Cl_{root}$        | $\Delta\%$                 |
| Drag coefficient at blade root                     | $Cd_{root}$        | $\Delta\%$                 |
| Leading edge separation time                       | uaTf0              | -                          |
| Leading edge pressure time                         | uaTp               | -                          |
| Trailing edge separation angle of attack           | aoaTES             | $\Delta\%$                 |
| Max lift angle of attack                           | aoaClMax           | $\Delta\%$                 |
| Separation reattachment angle of attack            | aoaSR              | $\Delta\%$                 |
| Yaw error                                          | Yaw                | degrees                    |
| Blade pitch error                                  | $\Theta_{blade}$   | degrees                    |
| Blade twist at tip                                 | $Twist_{tip}$      | degrees                    |
| Platform alignment                                 | $\Theta_{ptfm}$    | degrees                    |
| Tower stiffness multiplier                         | TK                 | -                          |
| Tower damping multiplier                           | TD                 | -                          |
| System horizontal center of gravity                | $COG_X$            | $\Delta\%$                 |
| System pitch and roll inertia                      | IYY                | $\Delta\%$                 |
| System mass                                        | Mass               | $\Delta\%$                 |
| Mooring line polyester length                      | $L_{fiber}$        | $\Delta\%$                 |
| Significant wave height                            | Hs                 | m                          |
| Peak wave period                                   | Tp                 | S                          |
| Wind-wave misalignment                             | $\Theta_{mis}$     | degrees                    |
| Current speed                                      | $V_{current}$      | m/s                        |
| Current direction                                  | $\Theta_{current}$ | degrees                    |
| Drag coefficient for upper column                  | $Cd_{upper}$       | -                          |
| Drag coefficient for lower column                  | $Cd_{lower}$       | -                          |
| Axial drag coefficient for outer columns           | $Cd_{axial}$       | -                          |
| Drag coefficient for rectangular pontoons          | $Cd_{rectangular}$ | -                          |
|                                                    |                    |                            |

The calculated sensitivities are directly proportional to the selected ranges for each variable input parameter. It is important to carefully consider what the true range of variability is for each input. All selected parameter ranges are listed in Table 5 for Humboldt Bay and in Table 6 for the Gulf of Maine. Parameter ranges that don't change between the two locations are only listed once in Table 5.

 Table 5. Parameter ranges for Humboldt Bay.

|                    | Minimum |                   |             | Nominal   |        |           |    |           |       | Maximum      |             |             |
|--------------------|---------|-------------------|-------------|-----------|--------|-----------|----|-----------|-------|--------------|-------------|-------------|
|                    | VI      | VR                | vo          | VI        | v      | /R        |    | vo        |       | VI           | VR          | vo          |
|                    | OP SU   | OP   FT   SU   SD | OP FT SU SD | OP SU     | -      | OP FT SU  | SD | OP FT     | SU SD | OP SU        | OP FT SU SD | OP FT SU SD |
| $U_{mean}$         | 3.00    | 8.59              | 23.00       | 3.00      | 10     | 0.59      |    | 25.00     |       | 5.00         | 12.59       | 25.00       |
| Shear              | -0.1206 | 0.0355            | 0.0826      | 0.0517    | 0      | 0.156     |    | 0.1285    |       | 0.3521       | 0.3641      | 0.1838      |
| $GM_{speed}$       | 0.00    |                   |             | 15.00 2.3 | 1   1: | 5.00 3.98 |    | 15.00 7.1 | 16    | 15.00   2.31 | 15.00 3.98  | 15.00 7.16  |
| $GM_{dir}$         | 0.0     |                   |             | 180.0     | 6      | 58.0      |    | 28.8      |       | 180.0        | 68.0        | 28.8        |
| $GM_{vshear}$      | 0.0     |                   |             | 4.67      | 6      | 5.25      |    | 9.25      |       | 4.67         | 6.25        | 9.25        |
| $GM_{hshear}$      | 0.0     |                   |             | 4.67      | 6      | 5.25      |    | 9.25      |       | 4.67         | 6.25        | 9.25        |
| $T_{Gust}$         | 6.0     |                   |             | 10.5      |        |           |    |           |       | 142.9        |             |             |
| $f_{return}$       | 0.0     |                   |             | 0.5       |        |           |    |           |       | 1.0          |             |             |
| $f_{dip}$          | 0.0     |                   |             | 0.5       |        |           |    |           |       | 1.0          |             |             |
| $Cl_{tip}$         | -26 %   |                   |             | 0 %       |        |           |    |           |       | 26 %         |             |             |
| $Cd_{tip}$         | -64 %   |                   |             | 0 %       |        |           |    |           |       | 64 %         |             |             |
| $Cl_{root}$        | -26 %   |                   |             | 0 %       |        |           |    |           |       | 26 %         |             |             |
| $Cd_{root}$        | -64 %   |                   |             | 0 %       |        |           |    |           |       | 64 %         |             |             |
| uaTf0              | 3.00    |                   |             | 6.50      |        |           |    |           |       | 10.00        |             |             |
| uaTp               | 1.00    |                   |             | 1.35      |        |           |    |           |       | 1.70         |             |             |
| aoaTES             | -0.20   |                   |             | 0.00      |        |           |    |           |       | 0.20         |             |             |
| aoaClMax           | -0.08   |                   |             | 0.00      |        |           |    |           |       | 0.08         |             |             |
| aoaSR              | -0.15   |                   |             | 0.00      |        |           |    |           |       | 0.15         |             |             |
| Yaw                | -8.0    |                   |             | 0.0       |        |           |    |           |       | 8.0          |             |             |
| $\Theta_{blade}$   | -2.0    |                   |             | 0.0       |        |           |    |           |       | 2.0          |             |             |
| $Twist_{tip}$      | -2.0    |                   |             | 0.0       |        |           |    |           |       | 2.0          |             |             |
| $\Theta_{ptfm}$    | 0.0     |                   |             | 0.0       |        |           |    |           |       | 60.0         |             |             |
| TK                 | 0.71    |                   |             | 0.00      |        |           |    |           |       | 1.22         |             |             |
| TD                 | 0.10    |                   |             | 0.00      |        |           |    |           |       | 5.00         |             |             |
| $COG_X$            | -2.0 %  |                   |             | 0.0 %     |        |           |    |           |       | 2.0 %        |             |             |
| IYY                | -2.0 %  |                   |             | 0.0 %     |        |           |    |           |       | 2.0 %        |             |             |
| Mass               | -2.0 %  |                   |             | 0.0 %     |        |           |    |           |       | 2.0 %        |             |             |
| $L_{fiber}$        | -2.5 %  |                   |             | 0.0 %     |        |           |    |           |       | 2.5 %        |             |             |
| Hs                 | 0.84    | 1.05              | 2.67        | 1.82      | 2      | 2.20      |    | 4.01      |       | 4.04         | 4.14        | 5.86        |
| Tp                 | f(Hs)   | f(Hs)             | f(Hs)       | f(Hs)     |        | (Hs)      |    | f(Hs)     |       | f(Hs)        | f(Hs)       | f(Hs)       |
| $\Theta_{mis}$     | 0.0     | 0.0               | 0.0         | 0.0       |        | 0.0       |    | 0.0       |       | 169.2        | 157.5       | 134.9       |
| $V_{current}$      | 0.06    | 0.07              | 0.12        | 0.22      | 0      | 0.26      |    | 0.34      |       | 0.53         | 0.61        | 0.69        |
| $\Theta_{current}$ | 0.0     |                   |             | 0.0       |        |           |    |           |       | 180.0        |             |             |
| $Cd_{upper}$       | 0.4     |                   |             | 1.3       |        |           |    |           |       | 2.0          |             |             |
| $Cd_{lower}$       | 0.4     |                   |             | 0.4       |        |           |    |           |       | 2.0          |             |             |
| $Cd_{axial}$       | 0.7     |                   |             | 8.0       |        |           |    |           |       | 10.0         |             |             |
| $Cd_{rectangular}$ | 0.7     |                   |             | 4.0       |        |           |    |           |       | 10.0         |             |             |

Table 6. Parameter ranges for Gulf of Maine (only including parameters with different ranges than shown for Humboldt Bay in Table 5).

|                | Minimum    |                            |                           | Nominal |             |             | Maximum |             |             |
|----------------|------------|----------------------------|---------------------------|---------|-------------|-------------|---------|-------------|-------------|
|                | VI         | VR                         | VO                        | VI      | VR          | VO          | VI      | VR          | vo          |
|                | OP SU      | OP FT SU SD                | OP   FT   SU   SD         | OP SU   | OP FT SU SD | OP FT SU SD | OP SU   | OP FT SU SD | OP FT SU SD |
| Shear          | -0.130     | 0.011                      | 0.049                     | 0.017   | 0.046       | 0.145       | 0.211   | 0.166       | 0.220       |
| Hs             | 0.40       | 0.52                       | 1.53                      | 0.80    | 1.24        | 2.95        | 1.85    | 2.52        | 5.65        |
| Tp             | f(Hs) wind | speed and site specific fu | unction shown in Figure 1 | 2       |             |             |         |             |             |
| $\Theta_{mis}$ | 0          | 0.0                        | 0.0                       | 0.0     | 0.0         | 0.0         | 176.5   | 172.8       | 90.3        |
| $V_{current}$  | 0.02       | 0.02                       | 0.04                      | 0.12    | 0.13        | 0.17        | 0.27    | 0.30        | 0.45        |

#### 4.1 Wind and aerodynamics

The ranges for the mean wind speed before the gust follow the IEC precedent in DLCs 1.4, 2.3, 3.2, 3.3, and 4.2 for a 2 m/s variation from the nominal value. This is the base hub height wind speed before the arrival of the gust. The range of shear profiles before the gust is selected based on the 2023 National Offshore Wind (NOW23) dataset which uses the Weather Research and Forecasting model tuned with data from offshore lidars, buoys and coastal radars (Bodini et al., 2023, 2024). The vertical wind speed distribution was fit to a power law function for each time in the data. Figure 8 shows the distribution of shear exponents that occurred with the relevant wind speeds for each selected wind speed range. The limits of the range for each location are set as the 10 % and 90 % quantiles of these conditional data.

Figure 8. Wind speed range conditional probability of shear exponents for Humboldt Bay and Gulf of Maine.

Gust magnitude values are able to range from 0.0 to a maximum of the IEC prescription. This leads to gusts tested that include only some elements of the combined gust, gusts that include all four changes, and gusts that include some combination of partial changes. This allows the assessment of the importance of each type of wind change within the gust. In IEC, a shear gust is only included in the EWS but is applied in the parametric gust where the maximum comes directly from the IEC function of rotor diameter, hub height, turbine class, and wind speed (IEC, 2019). Speed changes are present in both the EOG

and the ECD, and direction changes are present in both the EDC and ECD. Speed changes in the EOG are a function of rotor diameter, hub height, turbine class, and wind speed, while the ECD speed change is constant. Direction change in the EDC is also a function of rotor diameter, hub height, turbine class, and wind speed, while the ECD direction change is a function of only wind speed (IEC, 2019). Figure 9 shows the prescribed speed and direction change magnitudes for the three gust types. The vertical dashed lines show the relevant nominal wind speeds for the load cases in this study.

**Figure 9.** Gust speed and direction change magnitudes prescribed by IEC for EOG, ECD, and EDC. Dashed lines indicate the nominal wind speeds for the considered load cases (IEC, 2024).

For all cases, the ECD has a significantly larger change than the EOG or EDC. The more extreme ECD value was selected for all load cases in the present study. However, the rate was limited to a maximum rate expected for a given load case type as defined by IEC, as described in Table 7. For example, if the speed rate of change in a fault case exceeds that based on an EOG, the gust magnitude for speed is limited to the value resulting in that maximum rate. For cases with longer gust periods, where the rate would not be exceeded, the magnitude can go all the way to the larger ECD value.

Table 7. Rate limits.

205

|           | Speed     |      | Direction |      | Shear     |      |
|-----------|-----------|------|-----------|------|-----------|------|
|           | Magnitude | Rate | Magnitude | Rate | Magnitude | Rate |
| Operating | ECD       |      | ECD       |      | EWS       |      |
| Fault     | ECD       | EOG  | ECD       | EDC  | EWS       |      |
| Startup   | ECD       | EOG  | ECD       | EDC  | EWS       |      |
| Shutdown  | ECD       | EOG  | ECD       | EDC  | EWS       |      |

The period of the gust has a minimum of the lowest IEC gust type period (which is a fast gust) and a maximum of the VolturnUS-S surge natural period because the IEC standard for floating offshore wind turbines recommends assessing whether gusts aligned with floater natural periods are important (IEC, 2024). As shown in Fig. 1, the return and dip factors for the gust profile can vary between 0 and 1.

The airfoil polars are parameterized into seven parameters, following precedent from previous airfoil sensitivity studies (Abdallah et al., 2015; Hu et al., 2025; Shaler et al., 2019; Robertson et al., 2019). Abdallah et al. (2015) explained that both the lift coefficient, Cl, and drag coefficient, Cd, have been found to have high correlation across the range of angles of attack.

Because of this, the lift and drag are scaled uniformly at all angles. A separate parameter is set for variations in Cl and Cd at the blade root and tip. The variation for mid-span sections are linearly interpolated between the tip and root parameters. The angle of attack for three characteristic flow phenomenon: trailing edge separation (aoaTES), maximum lift (aoaClMax), and separation reattachment (aoaSR), are also included as variable control points. aoaTES is defined as the end of the linear region, and aoaSR is defined as the local minimum after maximum lift if it exists, or the maximum lift slope in that region if it does not. Changes in angle of attack are held constant below the start of the linear lift region (aoaLin) and above the point of zero lift (aoaCl0). Abdallah et al. (2015) found the correlation between these points to be weaker, justifying independent variations. Ranges for these seven parameters are taken as plus or minus 2 standard deviations based on experimental ranges of uncertainty due to variations in measurement, 3D rotational corrections, surface roughness, geometric distortions, Reynolds number, post-stall extensions, and full-scale field comparisons (Abdallah et al., 2015). Figure 10 shows the possible distortion of the lift and drag curves for a representative section of the IEA 15 MW reference wind turbine blade. The lift control point angles of attack are shown with markers. The blue curve shows the original function, and the cyan curves show the minimum and maximum variations possible for each parameter. The same is shown in red and pink for the drag curve.

**Figure 10.** Limits of perturbations for lift coefficient, trailing edge separation angle of attack, maximum lift angle of attack, separation reattachment angle of attack, and drag coefficient. Control point angles of attack are marked and labeled in the legend. The base lift curve is shown in blue, and the base drag curve is shown in red. Resulting perturbed lift curves from minimum and maximum values are shown in cyan. Resulting perturbed drag curves from minimum and maximum values are shown in pink.

Two unsteady aerodynamic model parameters are also included in the study. These include the two time constants used by the Beddoes–Leishman (BL) Hansen–Guanaa–Madsen (HGM) four-states model (Hansen et al., 2004). These parameters were previously included in sensitivity analyses focused on normal operating conditions, and loads were found to not be highly sensitive to them. However, it was thought that the importance of the parameters could be greater for the transient conditions in gusts. Ranges were set that matched those of the normal operating condition study (Shaler et al., 2019). The previous work used the BL Minnema–Pierce model, which includes two additional time constants and the Strouhal number, and included these as variable input parameters, but they are not inputs to the HGM model used in the present study. The ranges cover the experimental values for three wind turbine airfoils, which are described in the implementation report for

https://doi.org/10.5194/wes-2025-228

Preprint. Discussion started: 14 November 2025

© Author(s) 2025. CC BY 4.0 License.

this model in OpenFAST (Damiani et al., 2016). A 2025 paper reports narrower ranges for these time constants based on uncertainty quantification analysis trained on a larger set of experimental data (Hu et al., 2025). It was decided to keep the more conservative larger range since these parameters were not previously flagged, and further investigation of their uncertainty could be performed if they showed up as leading to key sensitivities. It should be noted that the BL HGM four-states model is not meant to handle deep stall and is set to turn off when angles of attack exceed 45 degrees, which is likely for some of the gust cases. It is also noted that other parameters used by the HGM model are based on the static polars, so variations in the static airfoil polars discussed above also affect the unsteady airfoil aerodynamics.

#### 4.2 System and structure

Ten parameters are included describing the turbine, structure, and mooring. Each of these parameters were highlighted as leading to some interesting level of sensitivity in the previous storm conditions EE sensitivity study with this turbine and platform. The ranges used match those from the previous work (Wiley et al., 2025).

Turbine parameters include yaw misalignment relative to the wind direction before the gust, blade pitch error, and blade twist error at the tip. Platform parameters include platform alignment relative to the wind direction before the gust; tower stiffness; tower damping; and the full system mass, horizontal center of gravity, and pitch and roll inertia. For the mooring lines, the variation in length of fiber sections due to creep was highlighted in the prior publication (Wiley et al., 2025).

#### 4.3 Water and hydrodynamics

Wave and current data came from the National Oceanic and Atmospheric Administration's National Data Buoy Center. Buoy number 46022 was used for Humboldt Bay data, and buoy numbers 44005 and 44032 were used for Gulf of Maine data. Detailed information about the data processing for these buoys can be found in the storm condition EE analysis paper (Wiley et al., 2025).

The significant wave height range was taken as the 5 % and 95 % quantiles of the data occurring within the relevant wind speed range for each load case. These ranges are shown with bold lines in Fig. 11.

**Figure 11.** Significant wave height as a function of hub height wind speed for Humboldt Bay and Gulf of Maine. The 5 %, 50 %, and 95 % quantiles are shown for each bin in red, blue, and red. Relevant wind speed ranges for selected load cases are marked with bold lines.

The peak period range was taken as a function of the significant wave height for every simulation. The red lines in Figure 12 show the 1 % and 99 % quantiles of peak period data, and the horizontal dashed lines show the possible range of significant wave heights associated with each wind speed. This range is determined for each case separately to avoid any non-physical waves.

**Figure 12.** Significant wave height and peak period contours for Humboldt Bay and Gulf of Maine. Red lines indicate the 1 % and 99 % quantiles. Dashed lines indicate the relevant significant wave heights for the three relevant wind speed ranges.

Wind-wave misalignment (before the gust) is similarly taken as the 1 % and 99 % quantiles of misalignment values occurring coincidentally with the relevant wind speeds. The full set of values and those lining up with the selected wind speed ranges are shown in Fig. 13.

Figure 13. Wind speed range conditional probability of wind-wave misalignment for Humboldt Bay and Gulf of Maine.

Current speed ranges are found with a similar 5 % and 95 % quantile of data coincident with the relevant wind speed ranges, as shown in Fig. 14.

**Figure 14.** Current speed as a function of hub-height wind speed for Humboldt Bay and Gulf of Maine. The 5 %, 50 %, and 95 % quantiles are shown for each bin in red, blue, and red. Relevant wind speed ranges for selected load cases marked with bold lines.

Platform drag coefficient ranges match those from previous studies with the same semisubmersible platform (Wiley et al., 2025; Sandua-Fernandez et al., 2022).

## 260 4.4 Quantities of interest

The EE sensitivity is assessed for each quantity of interest (QOI) independently. The selected QOI are given in Table 8. Each of these loads are considered for all load case types and wind speeds, and they are only assessed for ultimate loads.

Table 8. Output quantities of interest for ultimate load assessment

| Output Quantities of Interest     | Units            |
|-----------------------------------|------------------|
| Root bending moment               | N-m              |
| Root pitching moment              | N-m              |
| Low-speed shaft bending moment    | N-m              |
| Yaw-bearing bending moment        | N-m              |
| Yaw-bearing yaw moment            | N-m              |
| Tower-base bending moment         | N-m              |
| Blade-tip out-of-plane deflection | m                |
| Fairlead tension                  | N                |
| Anchor tension                    | N                |
| Watch circle                      | m                |
| Heel angle                        | degrees          |
| Nacelle acceleration              | m/s <sup>2</sup> |
| Power                             | W                |
|                                   |                  |

#### 5 Results

A total of 510,720 simulations were set up and post-processed for the analysis (2 locations, 10 load cases, 32 starting points, 37+1 parameters, 21 seed numbers). The resulting sensitivities are shown in Fig. 15 for Humboldt Bay and Fig. 16 for Gulf of Maine. The number of significant EE events occurring with a perturbation to a certain input parameter indicates the relative sensitivity to that input. In the plots, the bar for each input parameter is broken by color and hatch type to indicate for which output the significant EE value was found. Rotor-specific loads have no hatch, and more global system outputs have a diagonal line hatch. The fully opaque bars at the right of each parameter include all load cases combined together. The four translucent bars to the left are the resulting sensitivities if only an individual load case type is included in the post-processing. The significant threshold is reset based only on the outputs from that load case type. It is then possible to see if certain parameters only drive loads depending on the turbine condition or associated parameter ranges.

For the deep-water taut mooring used in Humboldt Bay, variations in the length of the fiber section of the mooring lines, which can vary over time due to creep, lead to the largest number of significant events. Most of these significant EE values are isolated to platform motions and mooring loads, with a small number of rotor-specific load sensitivities. It is expected that if internal reaction loads within the floater (which were not included in the present study) were calculated, these would also be strongly correlated to mooring loads. The fiber length variable leads to about 4 times fewer significant events for the semi-taut intermediate depth mooring system in the Gulf of Maine. This specific sensitivity to the polyester mooring line for a deep-water taut configuration was also found for extreme storm conditions in a previous study (Wiley et al., 2025). For most other input parameters, the relative sensitivities were generally similar for the two locations.

Figure 15. Ultimate load EE sensitivity for Humboldt Bay

For both locations, there are important sensitivities to parameters defining the gust. The direction change magnitude in the gust and the period of the gust are particularly important. The change in speed also triggers a relatively large number of significant events while changes to shear in both directions are less impactful. In terms of the gust shape, the dip appears to have a much larger influence than the return. Differences in the sensitivity to the inflow wind parameters between the two locations are largely driven by mooring loads and platform displacements. In deep-water Humboldt Bay, changes to the mooring line length change the significant threshold for these loads enough that the inflow parameters are far less dominant, whereas in the Gulf of Maine, the starting wind speed, gust speed magnitude, gust direction magnitude, and gust period contribute more significantly to platform motion loads.

Figure 16. Ultimate load EE sensitivity for Gulf of Maine

Relatively similar sensitivity levels are found for each of the selected aerodynamic parameters. This is in contrast to previous findings in operational conditions, where dynamic stall parameters showed negligible influence (Shaler et al., 2019). The two dynamic stall time constants show almost no significant impact in the startup load case type, and in Humboldt Bay, significant events are almost exclusive to the operational load case type. Across all aerodynamic parameters, only the angle of attack of trailing-edge separation (the point where lift is no longer linear with respect to angle of attack) has disproportionate importance in the startup cases. In these load cases, two parameters related to the geometric blade angle, blade pitch error and twist at the tip, also lead to a disproportionate number of significant EE values.

In the Gulf of Maine, where mooring load and platform watch circle sensitivities are not masked by the dependence on fiber length, the platform orientation is also a large driver for these outputs. For both locations, the platform horizontal center of gravity is the primary driving factor for heel angle. In some load case types, this also translates to the largest tower-base bending moment sensitivity.

For some load case types (startup and shutdown for Humboldt Bay and startup for Gulf of Maine), the three parameters describing the wave conditions have the strongest impact on nacelle accelerations. This is not the case when all load cases are included, and all water environment and hydrodynamic parameters lead to a similar number of significant events across output types.

For both locations, but especially for Humboldt Bay, the results when including all load case types strongly reflect the results based only on the operating load case. The inclusion of the nominal value in Eq. (2) allows one load case type and wind speed combination to dominate the sensitives, if it results in much larger loads. This can be seen in Fig. 17. For each QOI the plot shows the number of significant events that were flagged for each of the 10 load cases when the threshold was set with all results included. The operating load case near cut-in is by far the case that triggers the largest number of significant events.

320

The extreme ECD gust conditions in this load case lead to large loads and changes in loads, moving the significant threshold to the point where other dependencies are masked. The dependence of mooring loads and watch circle on fiber length and the heel angle on horizontal center of gravity are less dependent on load case. While the operating at cut-in load case still provides many significant events, it is less dominant in the Gulf of Maine results. It is interesting that this is true across most QOI, not just for the loads most closely linked to the different mooring system.

Figure 17. Number of significant events occurring in each load case type and wind speed

It should be noted that the near cut-in wind speed was included for the operating load case type since IEC DLC 1.5 (extreme wind shear) covers the full range from cut-in to cut-out. However, the large direction and wind speed changes of the ECD are only prescribed near rated speed in IEC DLC 1.4. While not applied in a DLC for all wind speeds, the gust magnitudes shown in Fig. 9 are given as a function of wind speed, with the intention of representing realistic gusts covering the full range of wind speeds (IEC, 2019). Even if the ECD is realistic near cut-in – and it is good to know that those loads drive ultimate load assessment for transient gust load cases – it can be easier to assess secondary levels of sensitivity when those data points are not included in the post-processing. Appendix A includes variations of Figs. 15, 16, and 17 but generated when all results from operating at cut-in are removed. While still important, the sensitivity to the magnitude of direction change drops significantly for both locations. Differences between the secondary sensitivities become more apparent, especially for Humboldt Bay, as shown in Fig. A2.

https://doi.org/10.5194/wes-2025-228 Preprint. Discussion started: 14 November 2025

© Author(s) 2025. CC BY 4.0 License.

The bar plot representation of the results provides an immediate visual understanding of the relative levels of sensitivity.

However, it can take some time to compare the contributions to significant events for a specific QOI. Results are also provided for both locations in Appendix B in a tabular format, which makes it easier to focus on a single QOI.

#### 6 Convergence

#### 6.1 Seed convergence

In simulations with a turbulent inflow, the random phasing applied to the frequency-dependent turbulent components is controlled with specified seed numbers to assess multiple random signals in a controlled way. In the load cases of this study, a deterministic gust is used so seed number does not change the inflow wind. A seed number is still used for the random phasing of the wave components, a random initial azimuth angle for the turbine, the positive/negative sign on the perturbation, and a random phase for the relative timing of the gust and a turbine event, as shown in Fig. 6. It is expected that without a random turbulent wind inflow, the results would be less dependent on the seed number, but it is still important to know that a sufficient number of seeds have been used so that differences in loads from perturbations in the input parameters are clearly distinguished from differences due to seed number. Figure 18 shows the total number of significant EE events associated with each input parameter (the total height of the bars in Figs. 15 and 16), calculated with an increasing number of seed numbers included in the dataset. A total of 21 seed numbers were used for all load cases. At this point, the relative sensitivity for the primary and secondary parameters of importance appears to be clear and not sensitive to additional seed numbers.

Figure 18. Convergence of ultimate load EE sensitivity for an increasing number of random seed numbers

Figures C1 and C2 in Appendix C show an additional confirmation of seed convergence focused on an individual input parameter and QOI combination example.

#### 6.2 Starting point convergence

Some input parameter/QOI sensitivities are expected to be nonlinear across the parameter range, and some sensitivities are expected to be the result of coupled relationships between multiple input parameters. Both of these possibilities result in EE values that vary across the parameter hyperspace; that is, depending on where the starting point is before perturbations, the calculated EE values will be different. The Sobol sequence is used to sample around the hyperspace, with an increasing number of starting points, until an approximation of global sensitivity is reached.

Figure 19 shows the proportion of significant events attributed to each input parameter as an increasing number of starting points is added to the sampling space. A total of 32 starting points were run for each load case in this study. As with the seed convergence, it appears that additional starting points will not result in any large changes to the relative sensitivity levels.

Figure 19. Convergence of ultimate load EE sensitivity for an increasing number of input parameter hyperspace starting points

Convergence of the final sensitivity levels with increasing starting points indicates that a sufficient sampling of the hyperspace has occurred to gain an understanding of the global sensitivity, but it is also interesting to know if these nonlinearities or coupled sensitivities exist. Given the radial one-at-a-time EE sampling, it is difficult to differentiate a nonlinear relationship from multiparameter coupling, but the standard deviation of the EE values across the starting points for a given input parameter and QOI indicates either of these. Figures D1 and D2 in Appendix D show the mean and standard deviation of each EE value broken up for the different load case type and wind speed combinations. If the mean is high and the standard deviation is low, the sensitivity is high and generally uniform across the parameter space. If the standard deviation is high compared to the mean, the sensitivity is either nonlinear or dependent on some other variable input parameter.

### 7 Full set of EE values

Figures 20 and 21 show the probability of exceedance for each QOI EE value, with a perturbation in each input parameter for each combination of load case type and wind speed. Different marker types are assigned to the five inputs leading to the largest EE values. Line and marker colors are based on the load case type. The vertical red lines indicate the threshold for a significant EE value, with all points to the right of the line adding to the tallies used to assess sensitivity. The range of deviation for each QOI is visible, as is the load grouping between types of load cases. Large outliers are present for some QOI. In Humboldt Bay, the clear separation in large EE values for the operating load case type is clear. The exception to this is for the mooring loads and watch circle, which clearly have larger values for perturbations in fiber length across all load cases.

**Figure 20.** Probability of exceedance of all EE values in Humboldt Bay. Unique marker types are set for the input parameters resulting in the five highest EE values across all load case types. The legend only shows the marker type that is constant across load case types for each QOI. A separate line is plotted for each load case type and wind speed combination for a total of 10 lines per input parameter. The vertical red line indicates the threshold for a significant EE value (with data from all load cases included), 2 standard deviations above the mean.

In the Gulf of Maine, a significant number of EE values above the threshold also occur in the fault and shutdown load cases.

**Figure 21.** Probability of exceedance of all EE values in Gulf of Maine. Unique marker types are set for the input parameters resulting in the five highest EE values across all load case types. The legend only shows the marker type that is constant across load case types for each QOI. A separate line is plotted for each load case type and wind speed combination for a total of 10 lines per input parameter. The vertical red line indicates the threshold for a significant EE value (with data from all load cases included), 2 standard deviations above the mean.

#### 8 Timing of ultimate loads

For the transient load cases, it is interesting to also know when the ultimate load occurs relative to the gust. When load cases include some kind of change in turbine operation, the ultimate load may occur because of the turbine event. With variations in event, gust period, and gust/event random phase, the time of maximum load for each QOI can change. Figures 22 and 23 show the cumulative density function of the probability that the maximum load has occurred at a certain time. The time is given as a multiple of the gust period. For most load types and load cases, the majority of the ultimate loads occur within the gust period. Some QOI maxima are more likely to occur before the arrival of the gust, including power, and others are more likely to occur in the time after the gust, including the yaw bearing yaw bending moment. The timing is the least linked to the onset of the gust for startup load cases.

**Figure 22.** Time of occurrence of ultimate load relative to the start of the gust, normalized by the gust period, for Humboldt Bay (CDF = cumulative density function)

**Figure 23.** Time of occurrence of ultimate load relative to the start of the gust, normalized by the gust period, for Gulf of Maine (CDF = cumulative density function)

For the majority of simulations, the controller's safety system shuts down the turbine with the onset of the gust. Figure 24 shows the proportion of runs that finished the modeled time still in an operating condition. As expected, no fault or shutdown load case types should be operating at the end of the run. For operating load case types, the automatic shutdown was triggered between 80 and 85 % of the time. For startup load case types, there were no runs where a successful startup occurred and continued operation through the duration of the gust.

Figure 24. Turbine operating or shutdown condition at end of simulation

#### 9 Conclusions

An EE sensitivity analysis method has been tested for transient load cases, including deterministic parametric gusts in combination with different turbine operation events. The four IEC defined gust types were combined into a single gust definition with variable magnitudes for each component of the gust and options for the shape and period of the gust profile. The change in direction and speed in the gust had a significantly larger impact on loads compared to changes in the shear. The period of the gust, whether or not there is a dip in the opposite direction at the start of the gust, and the wind speed before the gust were also identified as important sensitivities to the inflow.

Variations in the horizontal center of gravity and length of the fiber section of mooring lines were found to dominate the changes to platform heel and platform translations and mooring loads, respectively. Two locations with different water depths and mooring systems were tested. The sensitivity to the fiber length was much stronger for the deep-water taut mooring system; for the semi-taut mooring system in intermediate water depth, the orientation of the platform relative to the wind also had a large impact on mooring loads and platform watch circle.

The focus of the study was ultimate loads, and IEC-recommended partial safety factors and the addition of nominal loads were used to assess loads across different load cases. Of the transient load cases considered, an operational turbine with a gust near cut-in wind speed triggered many of the significant EE contributions. The ECD type gust parameter ranges applied in this load case led to extreme loads, to the point where other dependencies were not highlighted. When this load case data were removed from the results, more nuanced relationships were visible in the secondary and tertiary levels of sensitivity; however, the primary drivers did not change.

At a high level across the two locations and load cases tested, the **primary** sensitivities included:

- Mean wind speed before gust
- Gust magnitude of direction change
- Gust period
- Platform alignment

https://doi.org/10.5194/wes-2025-228 Preprint. Discussion started: 14 November 2025 © Author(s) 2025. CC BY 4.0 License.

- System horizontal center of gravity
  - Mooring line polyester length

The **secondary** sensitivities included:

- Gust magnitude of speed change
- Gust shape initial dip in opposite direction
- Trailing-edge separation angle of attack
  - Blade pitch error
  - Blade twist at tip

The outcome of this research can help inform probabilistic design approaches, improve site suitability analyses, aid in the development of surrogate models, and inform propagation of uncertainty to support model validation. The deterministic DLCs prescribed by IEC could be reconsidered based on these results, especially if similar trends are found important for other wind turbines. Further validation of the probability of an ECD gust type appears to be particularly important.

The screening method demonstrated in this modeling helps to identify the input parameters most strongly impacting different loads; for the variables identified, future work could be done to better understand the level of uncertainty in these values. Similar future studies could also include gusts with large changes in veer, which is not currently included as an IEC gust type. Given the majority of simulations that ended with a shutdown turbine, further investigation into controller and safety systems would be valuable when considering these gust load cases. The large changes in angle of attack that occur in these load cases challenge the available dynamic stall models and lead to large blade deflections that challenge the structural model. Further work could include higher-fidelity models to check the possible error and any resulting changes in sensitivity. The EE sensitivity method has now been tested individually for most IEC DLC types; future work could look across all IEC DLCs in a single study to understand the most important parameters. Comparisons of the resulting sensitivities for varying rotor sizes and floater types would also provide meaningful insights.

Code availability. The base OpenFAST input files and open-format versions of figures will be publicly available on Zenodo at: XXXXXXX.

Appendix A: Results without operational load case near cut-in wind speed

Figure A1. Number of significant events occurring in each load case type and wind speed when the operating load case near cut-in wind speed is not considered

Figure A2. Ultimate load EE sensitivity for Humboldt Bay when the operating load case near cut-in wind speed is not considered

Figure A3. Ultimate load EE sensitivity for Gulf of Maine when the operating load case near cut-in wind speed is not considered

## Appendix B: Sensitivity tables

|                           |                  |              |           |           |           |              | ٥.       |           |                |                  |          |       |             |
|---------------------------|------------------|--------------|-----------|-----------|-----------|--------------|----------|-----------|----------------|------------------|----------|-------|-------------|
|                           |                  |              |           | /         | /5:       | d Berding    | MOTT.    | /         | /              | /                | /        | /     | /           |
|                           |                  | , Bending to | port 5    | hat Berdi | IG MOTT   | Berry        | M MOTT.  | TIP Defte | ction /        | , / <sub>3</sub> | . /      |       | //.         |
|                           |                  | Bending      | Ditching. | aft Beril | Jan-Deat. | dearing that | daze Mor | TIR DEL   | ction derision | not Tension      | in Chick | Angle | ile Accel   |
|                           | 6 <sub>0</sub> c | , 60g        | 5/5       | 104       | 194       | THE THE      | Hase Has | e TR Lair | er bu          | VASA MAY         | d. Keg   | Mar   | elle Accele |
| U <sub>mean</sub>         | 0.09             | 0.06         | 0.07      | 0.02      | 0.04      | 0.09         | 0.09     | 0.01      | 0.01           | 0.02             | 0.07     | 0.04  | 0.00        |
| Shear                     | 0.02             | 0.03         | 0.04      | 0.01      | 0.04      | 0.01         | 0.02     | 0.00      | 0.00           | 0.00             | 0.01     | 0.03  | 0.00        |
| GM <sub>speed</sub>       | 0.05             | 0.04         | 0.11      | 0.02      | 0.07      | 0.07         | 0.05     | 0.00      | 0.00           | 0.00             | 0.07     | 0.05  | 0.00        |
| GM <sub>dir</sub>         | 0.04             | 0.03         | 0.05      | 0.05      | 0.05      | 0.05         | 0.02     | 0.00      | 0.00           | 0.00             | 0.06     | 0.04  | 0.00        |
| GM <sub>vshear</sub>      | 0.00             | 0.00         | 0.01      | 0.01      | 0.01      | 0.00         | 0.00     | 0.00      | 0.00           | 0.00             | 0.00     | 0.02  | 0.00        |
| GM <sub>hshear</sub>      | 0.00             | 0.00         | 0.01      | 0.01      | 0.01      | 0.00         | 0.00     | 0.00      | 0.00           | 0.00             | 0.00     | 0.01  | 0.00        |
| $T_{Gust}$                | 0.06             | 0.12         | 0.10      | 0.02      | 0.11      | 0.08         | 0.07     | 0.00      | 0.00           | 0.00             | 0.10     | 0.14  | 0.00        |
| f <sub>return</sub>       | 0.00             | 0.01         | 0.02      | 0.03      | 0.01      | 0.03         | 0.00     | 0.00      | 0.00           | 0.00             | 0.03     | 0.02  | 0.00        |
| f <sub>dip</sub>          | 0.04             | 0.02         | 0.06      | 0.02      | 0.05      | 0.05         | 0.04     | 0.00      | 0.00           | 0.00             | 0.04     | 0.04  | 0.00        |
| Cltip                     | 0.05             | 0.04         | 0.05      | 0.01      | 0.02      | 0.00         | 0.03     | 0.00      | 0.00           | 0.00             | 0.00     | 0.01  | 0.00        |
| Cd <sub>tip</sub>         | 0.02             | 0.04         | 0.02      | 0.01      | 0.01      | 0.00         | 0.01     | 0.00      | 0.00           | 0.00             | 0.00     | 0.01  | 0.00        |
| Cl <sub>root</sub>        | 0.02             | 0.04         | 0.04      | 0.01      | 0.02      | 0.00         | 0.01     | 0.00      | 0.00           | 0.00             | 0.00     | 0.01  | 0.00        |
| Cd <sub>root</sub>        | 0.02             | 0.02         | 0.02      | 0.01      | 0.01      | 0.00         | 0.01     | 0.00      | 0.00           | 0.00             | 0.00     | 0.01  | 0.00        |
| uaTf0                     | 0.02             | 0.01         | 0.01      | 0.00      | 0.01      | 0.00         | 0.00     | 0.00      | 0.00           | 0.00             | 0.00     | 0.00  | 0.00        |
| uaTV0                     | 0.00             | 0.00         | 0.00      | 0.00      | 0.00      | 0.00         | 0.00     | 0.00      | 0.00           | 0.00             | 0.00     | 0.00  | 0.00        |
| uaTp                      | 0.01             | 0.02         | 0.02      | 0.00      | 0.01      | 0.00         | 0.00     | 0.00      | 0.00           | 0.00             | 0.00     | 0.00  | 0.00        |
| uaTVL                     | 0.00             | 0.00         | 0.00      | 0.00      | 0.00      | 0.00         | 0.00     | 0.00      | 0.00           | 0.00             | 0.00     | 0.00  | 0.00        |
| uaSt                      | 0.00             | 0.00         | 0.00      | 0.00      | 0.00      | 0.00         | 0.00     | 0.00      | 0.00           | 0.00             | 0.00     | 0.00  | 0.00        |
| aoaTES                    | 0.03             | 0.04         | 0.04      | 0.02      | 0.03      | 0.01         | 0.01     | 0.00      | 0.00           | 0.00             | 0.01     | 0.02  | 0.00        |
| aoaClMax                  | 0.02             | 0.01         | 0.01      | 0.00      | 0.00      | 0.00         | 0.00     | 0.00      | 0.00           | 0.00             | 0.00     | 0.01  | 0.00        |
| aoaSR                     | 0.02             | 0.01         | 0.01      | 0.00      | 0.00      | 0.00         | 0.01     | 0.00      | 0.00           | 0.00             | 0.00     | 0.01  | 0.00        |
| Yaw                       | 0.01             | 0.00         | 0.01      | 0.01      | 0.00      | 0.01         | 0.00     | 0.00      | 0.00           | 0.00             | 0.01     | 0.01  | 0.00        |
| ⊖ <sub>blade</sub>        | 0.05             | 0.04         | 0.05      | 0.03      | 0.04      | 0.02         | 0.03     | 0.00      | 0.00           | 0.00             | 0.02     | 0.03  | 0.00        |
| Twist <sub>tip</sub>      | 0.05             | 0.05         | 0.04      | 0.02      | 0.05      | 0.02         | 0.03     | 0.00      | 0.00           | 0.00             | 0.02     | 0.03  | 0.00        |
| Θ <sub>ptfm</sub>         | 0.02             | 0.02         | 0.02      | 0.01      | 0.01      | 0.01         | 0.01     | 0.01      | 0.01           | 0.04             | 0.01     | 0.02  | 0.00        |
| TK                        | 0.02             | 0.02         | 0.01      | 0.00      | 0.01      | 0.00         | 0.00     | 0.00      | 0.00           | 0.00             | 0.00     | 0.00  | 0.00        |
| TD                        | 0.02             | 0.01         | 0.01      | 0.00      | 0.00      | 0.00         | 0.00     | 0.00      | 0.00           | 0.00             | 0.00     | 0.00  | 0.00        |
| COG <sub>X</sub>          | 0.02             | 0.02         | 0.03      | 0.02      | 0.02      | 0.17         | 0.00     | 0.00      | 0.00           | 0.00             | 0.44     | 0.01  | 0.00        |
| IYY                       | 0.02             | 0.02         | 0.01      | 0.00      | 0.01      | 0.00         | 0.00     | 0.00      | 0.00           | 0.00             | 0.00     | 0.00  | 0.00        |
| Mass                      | 0.02             | 0.02         | 0.01      | 0.00      | 0.01      | 0.00         | 0.01     | 0.00      | 0.00           | 0.01             | 0.00     | 0.00  | 0.00        |
| Lfiber                    | 0.02             | 0.02         | 0.02      | 0.02      | 0.02      | 0.02         | 0.01     | 0.51      | 0.51           | 0.29             | 0.02     | 0.02  | 0.00        |
| Hs                        | 0.02             | 0.02         | 0.01      | 0.01      | 0.01      | 0.00         | 0.00     | 0.00      | 0.00           | 0.00             | 0.00     | 0.06  | 0.00        |
| Тр                        | 0.02             | 0.02         | 0.02      | 0.01      | 0.01      | 0.01         | 0.01     | 0.00      | 0.00           | 0.00             | 0.01     | 0.15  | 0.00        |
| Θ <sub>mis</sub>          | 0.02             | 0.02         | 0.02      | 0.01      | 0.02      | 0.02         | 0.01     | 0.00      | 0.00           | 0.00             | 0.00     | 0.11  | 0.00        |
| V <sub>current</sub>      | 0.02             | 0.01         | 0.02      | 0.01      | 0.01      | 0.00         | 0.01     | 0.00      | 0.00           | 0.01             | 0.00     | 0.01  | 0.00        |
| Θ <sub>current</sub>      | 0.02             | 0.01         | 0.01      | 0.02      | 0.01      | 0.01         | 0.00     | 0.00      | 0.00           | 0.03             | 0.01     | 0.01  | 0.00        |
| Cd <sub>upper</sub>       | 0.02             | 0.02         | 0.02      | 0.00      | 0.01      | 0.00         | 0.01     | 0.00      | 0.00           | 0.00             | 0.00     | 0.01  | 0.00        |
| Cd <sub>lower</sub>       | 0.02             | 0.01         | 0.01      | 0.00      | 0.01      | 0.00         | 0.00     | 0.00      | 0.00           | 0.00             | 0.00     | 0.01  | 0.00        |
| Cd <sub>axial</sub>       | 0.02             | 0.02         | 0.01      | 0.01      | 0.01      | 0.00         | 0.00     | 0.00      | 0.00           | 0.00             | 0.00     | 0.00  | 0.00        |
| Cd <sub>rectangular</sub> | 0.01             | 0.01         | 0.01      | 0.01      | 0.01      | 0.00         | 0.00     | 0.00      | 0.00           | 0.00             | 0.00     | 0.00  | 0.00        |

**Figure B1.** Fraction of all perturbations of a given input parameter that exceed the significant EE threshold for a given QOI for all load cases in Humboldt Bay

|                           |      |             |          | /         | /        | / 3        | Morn.   | /           | /          | /         | /         | /          | /          |
|---------------------------|------|-------------|----------|-----------|----------|------------|---------|-------------|------------|-----------|-----------|------------|------------|
|                           |      |             | atr. /   | <u> </u>  | MOTT.    | g Bending  | MOTT.   | / /         | rion /     | / /       | / /       | / /        | / /        |
|                           |      | , Bending h | port 5   | har Bendi | dan Dear | db da      | , / .or | . Dette     | ctio ensid | n Tensior | in Circle | /.e        | /.¿è       |
|                           | , gè | Beru        | Street C | Shaft L   | DW. L    | Dearing to | 25/     | SE TIP VOIN | ead to     | 701 N     |           | Arigle Wat | elle Accel |
| U <sub>mean</sub>         | 0.02 | 0.01        | 0.03     | 0.01      | 0.02     | 0.04       | 0.10    | 0.09        | 0.09       | 0.07      | 0.05      | 0.01       | 0.00       |
| Shear                     | 0.01 | 0.01        | 0.01     | 0.01      | 0.01     | 0.01       | 0.01    | 0.01        | 0.01       | 0.01      | 0.02      | 0.01       | 0.00       |
| GM <sub>speed</sub>       | 0.01 | 0.00        | 0.04     | 0.01      | 0.01     | 0.04       | 0.07    | 0.01        | 0.01       | 0.00      | 0.06      | 0.00       | 0.00       |
| GM <sub>clir</sub>        | 0.01 | 0.01        | 0.02     | 0.02      | 0.01     | 0.05       | 0.02    | 0.01        | 0.01       | 0.00      | 0.05      | 0.01       | 0.00       |
| GM <sub>vshear</sub>      | 0.01 | 0.01        | 0.01     | 0.01      | 0.01     | 0.01       | 0.00    | 0.00        | 0.00       | 0.00      | 0.00      | 0.01       | 0.00       |
| GM <sub>hshear</sub>      | 0.01 | 0.01        | 0.01     | 0.01      | 0.01     | 0.01       | 0.01    | 0.00        | 0.00       | 0.00      | 0.00      | 0.01       | 0.00       |
| T <sub>Gust</sub>         | 0.02 | 0.01        | 0.04     | 0.01      | 0.01     | 0.06       | 0.08    | 0.02        | 0.02       | 0.02      | 0.09      | 0.01       | 0.00       |
| f <sub>return</sub>       | 0.01 | 0.01        | 0.01     | 0.00      | 0.01     | 0.04       | 0.00    | 0.01        | 0.01       | 0.01      | 0.03      | 0.01       | 0.00       |
| f <sub>dip</sub>          | 0.02 | 0.01        | 0.03     | 0.02      | 0.02     | 0.04       | 0.04    | 0.01        | 0.01       | 0.01      | 0.06      | 0.01       | 0.00       |
| Cl <sub>tip</sub>         | 0.01 | 0.01        | 0.02     | 0.01      | 0.01     | 0.01       | 0.03    | 0.01        | 0.01       | 0.01      | 0.01      | 0.01       | 0.00       |
| Cd <sub>tip</sub>         | 0.01 | 0.00        | 0.01     | 0.00      | 0.01     | 0.00       | 0.01    | 0.01        | 0.01       | 0.01      | 0.01      | 0.00       | 0.00       |
| Cl <sub>root</sub>        | 0.01 | 0.01        | 0.01     | 0.01      | 0.01     | 0.01       | 0.01    | 0.01        | 0.01       | 0.01      | 0.00      | 0.01       | 0.00       |
| Cd <sub>root</sub>        | 0.01 | 0.01        | 0.01     | 0.01      | 0.01     | 0.01       | 0.01    | 0.01        | 0.01       | 0.01      | 0.01      | 0.01       | 0.00       |
| uaTf0                     | 0.01 | 0.01        | 0.01     | 0.01      | 0.01     | 0.01       | 0.01    | 0.01        | 0.01       | 0.01      | 0.01      | 0.01       | 0.00       |
| uaTV0                     | 0.00 | 0.00        | 0.00     | 0.00      | 0.00     | 0.00       | 0.00    | 0.00        | 0.00       | 0.00      | 0.00      | 0.00       | 0.00       |
| иаТр                      | 0.00 | 0.00        | 0.01     | 0.00      | 0.01     | 0.01       | 0.01    | 0.01        | 0.01       | 0.01      | 0.01      | 0.00       | 0.00       |
| uaTVL                     | 0.00 | 0.00        | 0.00     | 0.00      | 0.00     | 0.00       | 0.00    | 0.00        | 0.00       | 0.00      | 0.00      | 0.00       | 0.00       |
| uaSt                      | 0.00 | 0.00        | 0.00     | 0.00      | 0.00     | 0.00       | 0.00    | 0.00        | 0.00       | 0.00      | 0.00      | 0.00       | 0.00       |
| aoaTES                    | 0.01 | 0.01        | 0.01     | 0.01      | 0.01     | 0.01       | 0.02    | 0.01        | 0.01       | 0.01      | 0.01      | 0.01       | 0.00       |
| aoaClMax                  | 0.01 | 0.01        | 0.01     | 0.01      | 0.01     | 0.01       | 0.00    | 0.01        | 0.01       | 0.00      | 0.00      | 0.01       | 0.00       |
| aoaSR                     | 0.01 | 0.01        | 0.01     | 0.01      | 0.01     | 0.01       | 0.01    | 0.01        | 0.01       | 0.01      | 0.00      | 0.01       | 0.00       |
| Yaw                       | 0.01 | 0.01        | 0.01     | 0.01      | 0.01     | 0.01       | 0.00    | 0.01        | 0.01       | 0.00      | 0.00      | 0.01       | 0.00       |
| $\Theta_{blade}$          | 0.01 | 0.01        | 0.03     | 0.02      | 0.02     | 0.02       | 0.04    | 0.01        | 0.01       | 0.00      | 0.03      | 0.01       | 0.00       |
| $Twist_{tip}$             | 0.01 | 0.01        | 0.03     | 0.02      | 0.01     | 0.02       | 0.04    | 0.01        | 0.01       | 0.01      | 0.03      | 0.01       | 0.00       |
| $\Theta_{ptfm}$           | 0.01 | 0.01        | 0.02     | 0.01      | 0.01     | 0.02       | 0.01    | 0.09        | 0.09       | 0.12      | 0.02      | 0.01       | 0.00       |
| TK                        | 0.01 | 0.01        | 0.02     | 0.01      | 0.01     | 0.01       | 0.01    | 0.01        | 0.01       | 0.01      | 0.01      | 0.01       | 0.00       |
| TD                        | 0.01 | 0.01        | 0.01     | 0.01      | 0.01     | 0.01       | 0.01    | 0.01        | 0.01       | 0.01      | 0.01      | 0.01       | 0.00       |
| COG <sub>X</sub>          | 0.01 | 0.01        | 0.01     | 0.01      | 0.01     | 0.03       | 0.01    | 0.01        | 0.01       | 0.01      | 0.46      | 0.01       | 0.00       |
| IYY                       | 0.01 | 0.01        | 0.01     | 0.01      | 0.01     | 0.01       | 0.01    | 0.00        | 0.00       | 0.00      | 0.01      | 0.01       | 0.00       |
| Mass                      | 0.01 | 0.01        | 0.01     | 0.01      | 0.01     | 0.01       | 0.01    | 0.00        | 0.00       | 0.00      | 0.00      | 0.01       | 0.00       |
| L <sub>fiber</sub>        | 0.01 | 0.01        | 0.01     | 0.01      | 0.01     | 0.01       | 0.01    | 0.06        | 0.05       | 0.15      | 0.01      | 0.01       | 0.00       |
| Hs                        | 0.01 | 0.01        | 0.01     | 0.01      | 0.01     | 0.01       | 0.01    | 0.01        | 0.01       | 0.01      | 0.01      | 0.01       | 0.00       |
| Тр                        | 0.01 | 0.01        | 0.01     | 0.01      | 0.01     | 0.01       | 0.01    | 0.01        | 0.02       | 0.02      | 0.01      | 0.01       | 0.00       |
| $\Theta_{mis}$            | 0.01 | 0.01        | 0.01     | 0.01      | 0.01     | 0.01       | 0.00    | 0.01        | 0.01       | 0.01      | 0.01      | 0.01       | 0.00       |
| V <sub>current</sub>      | 0.01 | 0.01        | 0.01     | 0.01      | 0.01     | 0.01       | 0.01    | 0.01        | 0.01       | 0.01      | 0.01      | 0.01       | 0.00       |
| Θ <sub>current</sub>      | 0.01 | 0.01        | 0.01     | 0.01      | 0.01     | 0.01       | 0.01    | 0.01        | 0.01       | 0.01      | 0.01      | 0.01       | 0.00       |
| Cd <sub>upper</sub>       | 0.01 | 0.01        | 0.01     | 0.01      | 0.01     | 0.01       | 0.01    | 0.00        | 0.00       | 0.00      | 0.01      | 0.01       | 0.00       |
| Cd <sub>lower</sub>       | 0.01 | 0.01        | 0.01     | 0.01      | 0.01     | 0.01       | 0.00    | 0.00        | 0.00       | 0.00      | 0.01      | 0.01       | 0.00       |
| Cd <sub>axial</sub>       | 0.01 | 0.01        | 0.01     | 0.01      | 0.01     | 0.01       | 0.01    | 0.01        | 0.01       | 0.01      | 0.01      | 0.01       | 0.00       |
| Cd <sub>rectangular</sub> | 0.01 | 0.01        | 0.01     | 0.02      | 0.01     | 0.01       | 0.01    | 0.01        | 0.01       | 0.01      | 0.01      | 0.01       | 0.00       |

**Figure B2.** Fraction of all perturbations of a given input parameter that exceed the significant EE threshold for a given QOI for all load cases in Gulf of Maine

## CC BY

#### 430 Appendix C: Seed convergence for single input parameter and QOI combination

**Figure C1.** Convergence of ultimate nacelle acceleration change due to a perturbation in the gust period for an increasing number of random seed numbers for Humboldt Bay

**Figure C2.** Convergence of ultimate nacelle acceleration change due to a perturbation in the gust period for an increasing number of random seed numbers for Gulf of Maine

#### Appendix D: Mean and standard deviation of EE values

**Figure D1.** Mean and standard deviation of all EE values for Humboldt Bay. Points are plotted individually for each load case type and wind speed, resulting in 10 points for each input parameter type. A large mean value indicates high sensitivity, and a large standard deviation indicates either coupling between variables or a nonlinear relationship within the selected parameter range.

**Figure D2.** Mean and standard deviation of all EE values for Gulf of Maine. Points are plotted individually for each load case type and wind speed, resulting in 10 points for each input parameter type. A large mean value indicates high sensitivity, and a large standard deviation indicates either coupling between variables or a nonlinear relationship within the selected parameter range.

Author contributions. The project was conceptualized by JJ and AR, who provided coordination and direction of all aspects. The input parameter data processing and numerical model execution was performed by WW. The draft text was prepared by WW with editing and review from JJ and AR

435 *Competing interests.* At least one of the (co-)authors is a member of the editorial board of *Wind Energy Science*. The authors have no other competing interests to declare.

Acknowledgements. The authors would like to thank Daniel Zalkind from NREL for setting up the custom controller used in this study.

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

© Author(s) 2025. CC BY 4.0 License.

- Shaler, K., Robertson, A., and Jonkman, J.: Sensitivity analysis of the effect of wind and wake characteristics on wind turbine loads in a small wind farm, Wind Energy Science, 8, https://doi.org/10.5194/wes-8-25-2023, 2023.
  - Sobol, I.: On the distribution of points in a cube and the approximate evaluation of integrals, USSR Computational Mathematics and Mathematical Physics, 7, 86–112, 1967.
  - Sørum, S., Katsikogiannis, G., Bachynski-Polić, E., Amdahl, J., Page, A., and Klinkvort, R.: Fatigue design sensitivities of large monopile offshore wind turbines, Wind Energy, 25, 1684–1709, https://doi.org/10.1002/we.2755, 2022.
- Wiley, W., Robertson, A., Jonkman, J., and Shaler, K.: Sensitivity analysis of numerical modeling input parameters on floating offshore wind turbine loads, Wind Energy Science, 8, https://doi.org/10.5194/wes-8-1575-2023, 2023.
  - Wiley, W., Robertson, A., and Jonkman, J.: Sensitivity analysis of numerical modeling input parameters on floating offshore wind turbine loads in extreme idling conditions, Wind Energy Science, 10, https://doi.org/10.5194/wes-10-941-2025, 2025.