# Peer review of "Sensitivity analysis of numerical modeling input parameters on wind turbine loads in deterministic transient load cases"

_Wind Energy Science, 2025_

## Referee Comment (RC1)

**Review of the paper "Sensitivity analysis of numerical modeling input parameters on wind turbine loads in deterministic transient load cases" by W. Wiley, J. Jonkman and A. Robertson**

**General comment**

The paper presents the results of an extensive simulation campaign focused on evaluating the impact of variations in model parameters on the design loads of a floating wind turbine. The paper is well written, and the Authors have done an excellent job summarizing the results and presenting them in a clear and understandable manner. The paper is suitable for publication but there is one important comment that Authors should address before the final acceptance of the manuscript.

**Important comment**

- There are notable similarities between this work and that of Bortolotti et al., WES 20219 (Bortolotti, P., Canet, H., Bottasso, C. L., and Loganathan, J., Performance of non-intrusive uncertainty quantification in the aeroservoelastic simulation of wind turbines, *Wind Energy Science*, 4, 397–406, 2019). The paper by Bortolotti at al. focused on a bottom-fixed turbine and adopted a different approach to evaluate the propagation of the uncertainties. Although these represent two strong differences, I encourage the Authors to discuss both similarities and differences in the introduction.

**Minor comments**

1. Fig. 3: Consider enlarging and improving the figure, especially given the two-column layout of the journal, which may reduce its size. Mooring lines are barely visible and power cables could be drawn in a different color.
2. Line 85: The authors correctly state that the controller structure and gains are perfectly known, making it unnecessary to include control parameters in the uncertainty analyses. However, measurements (used by controllers) could be affected by systematic/deterministic errors that are not known a priori. Additionally, actuator bandwidth (and dynamics) could also be uncertain. This aspect should be mentioned, as they potentially affect the turbine response especially in faulty of gusty conditions.
3. Lines 146-152: the discussion is correct, but it is worth noting that if the range of each parameter is chosen based on its physical variability (or its standard deviation) then, the sensitivity may provide an intuitive piece of information. For example, for a single parameter the absolute value of the product between the sensitivity and the parameter standard deviation represents a good estimate of the output standard deviation. This addresses the issue of comparing sensitivities, mentioned by the authors.
4. Eq. 3: the meanings of $\sigma$ and $\mu$ are missing and should be easily introduced in the sentence immediately preceding the equation.

5. Tab. 4: some parameters, such as the lift and drag coefficient at blade tip, depend on the angle of attack. Please, provide a better explanation. Line 206 provides a partial clarification.
6. Lines 222-223: the inclusion of the unsteady aerodynamics parameters in the analysis is appreciated. It should also be noted that these parameters may influence not only the transient behavior but also the steady response under very large yaw misalignment angles (e.g. following the end of the ECD gust).
7. Fig. 20 and 21: it seems that the Gulf of Maine case appears to exhibit a generally lower probability of exceedance compared to Humboldt Bay. Is there a physical reason for this difference?

---

## Author Comment (AC1)

Dear referees,

Thank you for your time providing feedback and suggestions for our paper. Please find our answers to your comments and the corresponding updates to the paper below. We hope to have addressed all of your comments, improving the manuscript. We will be happy to have continued discussion. We will submit the revised manuscript including a pdf highlighting the differences made to the text in the next few days.

Thank you,

Will Wiley, Jason Jonkman, and Amy Robertson

**Referee 1**

*General comment: The paper presents the results of an extensive simulation campaign focused on evaluating the impact of variations in model parameters on the design loads of a floating wind turbine. The paper is well written, and the Authors have done an excellent job summarizing the results and presenting them in a clear and understandable manner. The paper is suitable for publication but there is one important comment that Authors should address before the final acceptance of the manuscript.*

*There are notable similarities between this work and that of Bortolotti et al., WES 2019 (Bortolotti, P., Canet, H., Bottasso, C. L., and Loganathan, J., Performance of non-intrusive uncertainty quantification in the aeroservoelastic simulation of wind turbines, Wind Energy Science, 4, 397–406, 2019). The paper by Bortolotti et al. focused on a bottom-fixed turbine and adopted a different approach to evaluate the propagation of the uncertainties. Although these represent two strong differences, I encourage the Authors to discuss both similarities and differences in the introduction.*

This is a good reference to mention in the introduction. The Bortolotti et al. paper considers a smaller number of uncertain input parameters (3 compared to 37) allowing the generation of surrogate models with two methods (non-intrusive polynomial chaos expansion and Universal Kriging) and comparisons to a Monte Carlo sampling. The use of a surrogate model could be an interesting addition for future work to gain more insights from the run simulations, but likely would have challenges with the discrete events considered in these specific load cases. The Bortolotti paper includes discussion of coupling effects which are not possible to distinguish with the one at a time method used in this paper, and require a prohibitively large number of simulations for a larger number of parameters.

Addition to Section 3, Paragraph 1: "Previous papers have included investigations of input parameter coupling impacts on wind turbine load uncertainty; the required number of simulations for this identification typically grows exponentially with the number of input parameters, becoming prohibitive for a large number of inputs. Techniques have also been demonstrated to use surrogate models to reduce computation cost (Bortolotti et al., 2019), but it is expected that these models would potentially have difficulties with the discrete events in the transient load cases. The one-at-a-time sampling with the EE method can only screen for the most influential input parameters, but with a required number of simulations that scales linearly with the number of inputs."

*Fig.3: Consider enlarging and improving the figure, especially given the two-column layout of the journal, which may reduce its size. Mooring lines are barely visible and power cables could be drawn in a different color.*

The figure came from a referenced paper, so we are not able to change the colors, but we have enlarged it and added more description of the power cable to the caption.

Increased figure to be full page width and added description of the power cable to the caption.

*L85: The authors correctly state that the controller structure and gains are perfectly known, making it unnecessary to include control parameters in the uncertainty analyses. However, measurements (used by controllers) could be affected by systematic/deterministic errors that are not known a priori. Additionally, actuator bandwidth (and dynamics) could also be uncertain. This aspect should be mentioned, as they potentially affect the turbine response especially in faulty of gusty conditions.*

Uncertainty in measurements used by the controller could be an interesting addition to future work. We did not have a clear expectation of what the potential range of this uncertainty could be to include in the current work. For the actuator bandwidth, sensitivity to the pitch and torque rates is shown uncoupled from all other variables in Figure 4. These rates were considered known and constant, but deviations from the expected chosen design values are possible due to manufacturing inconsistencies or aging components, and could be interesting to include in future work. We did not have a good estimation of the possible range, and chose not to include these variables in the current work.

Addition to Section 2.2, Paragraph 3: "Some error in sensor information that the controller relies on, and some changes to actuator bandwidth limits as components age, are possible. Ranges of variability for these parameters were not clear, but could be included in future studies."

*L146-152: The discussion is correct, but it is worth noting that if the range of each parameter is chosen based on its physical variability (or its standard deviation) then, the sensitivity may provide an intuitive piece of information. For example, for a single parameter the absolute value of the product between the sensitivity and the parameter standard deviation represents a good estimate of the output standard deviation. This addresses the issue of comparing sensitivities, mentioned by the authors.*

We wanted to emphasize that it is important to pick a reasonable parameter range, but it is a good point to also emphasize that if the range is well-selected, it is useful that the sensitivity depends on the range.

Addition to Section 3, Paragraph 5: "A reasonable range needs to be selected, but this dependence is useful in evaluating the possible uncertainty in loads. If an output is sensitive to a parameter, but the possible range of variability in that parameter is very small, the uncertainty in the load is also small."

*Eq.3: The meanings of $\sigma$ and $\mu$ are missing and should be easily introduced in the sentence immediately preceding the equation.*

The symbol definition has been added to the sentence preceding Eq. 3.

Section 3, Paragraph 6: "This threshold is defined in Eq. (3), as any EE value that is greater than 2 standard deviations, σ, above the mean, μ, of all EE values."

*Tab.4: Some parameters, such as the lift and drag coefficient at blade tip, depend on the angle of attack. Please, provide a better explanation. Line 206 provides a partial clarification.*

A clearer description of the variation to lift and drag was added.

Addition to Section 4.1, Paragraph 5: "Because of this, the lift and drag are scaled uniformly, with a certain percent change from the baseline polar applied across all angles of attack."

*L222-223: The inclusion of the unsteady aerodynamics parameters in the analysis is appreciated. It should also be noted that these parameters may influence not only the transient behavior but also the steady response under very large yaw misalignment angles (e.g. following the end of the ECD gust).*
This is a good point and was added to the text.
Addition to Section 4.1, Paragraph 6: "However, it was thought that the importance of the parameters could be greater for the transient conditions in gusts or the possibly large yaw misalignment angles resulting from the gusts"

*Fig. 20-21: It seems that the Gulf of Maine case appears to exhibit a generally lower probability of exceedance compared to Humboldt Bay. Is there a physical reason for this difference?*
One key difference between the two locations is the semi-taut mooring system in the Gulf of Maine and the taut mooring system in Humboldt Bay. For the taut mooring system, the sensitivity to the length of the fiber section, which has variability due to creep, dominates variability in the mooring loads and platform displacement. This explains the larger probability of exceedance for fairlead tension, anchor tension, and watch circle in Humboldt Bay. For other outputs, there is another interesting difference in the results for the two locations. In Humboldt Bay, the large majority of significant events come from the operating load case that includes the more extreme ECD gust conditions. The magnitude of the direction change, as well as the shape and period of the gust in combination with this large direction change have particularly large impacts on the loads. In the Gulf of Maine outputs, these parameters are relatively less impactful, so that other load case types are more likely to result in a significant output. The ranges for these gust parameters are the same for the two locations. It appears that all loads, including at the blade level, are more sensitive to changes in direction for the deep water system, which is most likely due to the difference in mooring system. Interestingly, in contrast to this hypothesis, the initial wind direction relative to the platform ($\theta_{ptfm}$) leads to more significant events in the Gulf of Maine.
Additions to Section 7, Paragraph 1: "In Humboldt Bay, the clear separation in large EE values for the operating load case type is clear. In this load case, the extreme direction change from the ECD gust type and the related shape and period gust parameters dominate the large changes to outputs. Interestingly, this is much less true for the Gulf of Maine, even though ranges for these parameters are the same for the two locations. The most significant difference between the two sites is the mooring system used for the different water depths. The exception to the operating load case type separation for Humboldt Bay is for the mooring loads and watch circle, which clearly have larger values for perturbations in fiber length across all load cases."

**Referee 2**

*General comment: An interesting article about the application of sensitivity analyses to transient load cases. As the framework has been used before, the novelty mainly lies in the application area to gust cases. The largest part of the paper is describing the methodology, which is quite rigorous and well written. Although there are many visualizations of the results, it is felt that a more elaborate interpretation of selected results (why are certain parameters more impactful?) could be valuable.*

Some added discussion / interpretation of the results has been added. We have tried to summarize the key findings, and we hope the different styles of results plots from Figures 15-D2 allow readers to further gain insights.

Additions to Section 7, Paragraph 1: "In Humboldt Bay, the clear separation in large EE values for the operating load case type is clear. In this load case, the extreme direction change from the ECD gust type and the related shape and period gust parameters dominate the large changes to outputs. Interestingly, this is much less true for the Gulf of Maine, even though ranges for these parameters are the same for the two locations. The most significant difference between the two sites is the mooring system used for the different water depths. The exception to the operating load case type separation for Humboldt Bay is for the mooring loads and watch circle, which clearly have larger values for perturbations in fiber length across all load cases."

Addition to Section 9, Paragraph 6: "A significant portion of the primary and secondary sensitivities are linked to parameters describing the gust. The standard deviation of a turbulent wind inflow was previously found to be the input parameter with the largest impact for operational load cases (Wiley et al., 2023). The results of the current study confirm the importance of understanding variations in the incident wind, even when a turbine is only operational for part of a load case. The extreme variations in speed and direction found in the IEC gust load cases can have large impacts on system loading, and emphasis should be placed on accurately characterizing these events."

*L11-13: Acknowledging the number of parameters to define the gusts in Table 4, the five impactful parameters mentioned in the abstract constitute almost three quarter of the total parameters. To what extent can this be considered as an expected outcome?*

Previous elementary effects sensitivity studies focused on floating offshore wind turbines found variations in inflow wind to be primary sensitivities for operational load cases, but not for extreme idling load cases (https://doi.org/10.5194/wes-8-1575-2023)(https://doi.org/10.5194/wes-10-941-2025). The current transient gust load cases include periods of turbine operation and idling, as well as wind conditions that could be considered extreme. It would be reasonable to expect that the significant changes to inflow wind would drive turbine loads, especially when the turbine is operating for some portion of the simulation. It is interesting that wave and current parameters appear to be much less important than for the fully idling extreme condition load cases. Given how important the gust parameters are to the current load cases, future work characterizing the uncertainty in gust magnitudes, shapes, and periods on the scales of large offshore wind rotors appears to be important.

Addition to Section 9, Paragraph 6: "A significant portion of the primary and secondary sensitivities are linked to parameters describing the gust. The standard deviation of a turbulent wind inflow was previously found to be the input parameter with the largest impact for operational load cases (Wiley et al., 2023). The results of the current study confirm the importance of understanding variations in the incident wind, even when a turbine is only operational for part of a load case. The extreme variations in speed and direction found in the IEC gust load cases can have large impacts on system loading, and emphasis should be placed on accurately characterizing these events."

*Tab.1: To what extent is the detail of this table needed for the storyline in section 2? Could it be removed as well and the relevant info explained in wording or is it relevant to refer to in the remainder of the paper?*

We believe it is important to have the context of the IEC load case types to justify why we set up the transient gust load cases in the way that we did. We think it is useful for the reader to be able to quickly

look back at this information as a reference when further along in the methodology and results section, and hope that a table makes this easier to do.

*Sec.2.2: A wind turbine on a semisubmersibe platform is chosen for this study instead of an onshore wind turbine. Is there a particular reason for this choice in the light of the gust cases? Would it be more straightforward to use a bottom fixed turbine for this purpose, yielding similar results for turbine load sensitivity to gusts?*

Studying the sensitivities within these load case types would also be interesting for an onshore turbine. Previous studies considering different IEC load cases focused on a floating system, and also incorporated input parameters related to the metocean conditions, floater, and mooring. We thought it would be interesting to also see how the relative significance of these floating specific parameters changed between different load case types. It would be simpler though to use a fixed or land based system.

Addition to Section 2.2, Paragraph 2: "Studying sensitivities in these transient load cases would also be interesting for a fixed bottom or land-based turbine, but using the same system from our prior studies offers the ability to compare sensitivities across different load case types."

*L124: Can the authors comment on the apparent absence of coupling between the parameters and its influence on the current sensitivity study?*

The elementary effects approach does not allow identification of coupling. The required number of simulations for coupling identification typically grows exponentially with the number of input parameters, making it prohibitive for the number of parameters of interest in this study and the computation time required for an aero-hydro-servo-elastic simulation. Elementary effects simulation requirements grow linearly with the number of parameters. Figures D1 and D2 show the mean and standard deviation of all EE values. The standard deviation quantifies how much the EE values change at different points in the parameter hyperspace. This variation could come from either coupling between parameters or nonlinearities in the relationship to that one parameter across the parameter range. The elementary effects technique works well as a screening method, to identify which parameters should be focused on in more detailed studies. With a smaller number of parameters identified from the screening method, a full variance based sensitivity analysis could be possible.

Addition to Section 3, Paragraph 1: "The one-at-a-time sampling with the EE method can only screen for the most influential input parameters, but with a required number of simulations that scales linearly with the number of inputs."

*L174: Please clarify based on what info and assumptions the different parameters for the two sites in table 5 and 6 are defined. If these are defined in the following sections it would be good to indicate that or place the tables afterwards.*

Reference is mentioned to the following sections just before the parameter range table.

Addition to Section 4, Paragraph 2: "Reasoning behind parameter range selection is described in Sections 4.1 – 4.3."

*L207: Acknowledging the differences in desired airfoil characteristics along the span of a wind turbine blade, can the authors comment on the validity of a linear interpolation of these characteristics between root and tip?*

Abdallah et al. discuss the correlation length of polar variability across the span of a blade (https://doi.org/10.1016/j.renene.2014.10.009). They determine that a correlation length of 15-20% of the blade span is likely. This length suggests that varying the polars uniformly over the span might be overly conservative, with larger than realistic changes to loads. Using separate variables for the tip and root is still likely a conservative approach, but limits the total number of parameters in the study. The lift and drag coefficients still do not show up as primary sensitivities for these load cases even with the conservative approach. If they did, more attention might need to be given to breaking up the variables further along the blade span.

Addition to Section 4.1, Paragraph 5: "This is likely a conservative approach, as, "the correlation length for the aerodynamic stochastic variables is of the order of 15%-20% of the blade length" (Abdallah et al., 2015)."

*Fig.15-16 & 18-21: Contain too many lines/bars and too small axis annotation and legend fonts for a reader to allow interpretation. Consider redesigning these plots to convey the necessary information in an adequate manner.*

When possible, figures were adjusted to increase font size for improved readability.

Figures 15, 16, A2, and A3: Moved legend below plot to increase font size.

Figures 18 and 19: Changed legend to use two columns to create room for larger font size.

*L421-423#: The aerodynamic model challenge mentioned relates to unsteady airfoil effects. It is known that the defined gusts also challenge dynamic wake and yaw models. Would this be a recommendation to investigate e.g. using a free vortex wake method?*

This is also a good consideration for a higher fidelity approach to model the challenging conditions with some increase in computational cost.

Addition to Section 9, Paragraph 8: "The large direction changes in the gusts also challenge the dynamic wake and yaw models; future use of the free vortex wake model available within OpenFAST, OLAF, could help address this issue with some added computational cost."